# A Control Method for the Differential Steering of Tracked Vehicles Driven Independently by a Dual Hydraulic Motor

**Jiangyi Han \*, Fan Wang**  **and Yuhang Wang**

School of Automotive and Traffic Engineering, Jiangsu University, Zhenjiang 212013, China;
wangfan9026@163.com (F.W.); 18845040396@163.com (Y.W.)
**\*** Correspondence: hjy0306@ujs.edu.com; Tel.: +86-188-0338-2330

**Abstract:** It is well known that tracked vehicles can adapt well to all kinds of terrain. However, the safety of tracked vehicles should be considered during steering on sloped terrain. This paper focuses on the differential steering control of tracked vehicles independently driven by a hydraulic motor. Firstly, the dynamic model of hydrostatic drive system was built and the kinematics and dynamics of differential steering driving were analyzed theoretically. Secondly, in order to prevent rollover of the tracked vehicle, the method of vehicle speed constraint was proposed. The constraint conditions of vehicle speed and steering angular velocity were analyzed under different slope conditions. Thirdly, based on the analysis results, differential steering control rules for tracked vehicles were formulated. To verify the effectiveness of the control rules, the models of vehicle driving dynamics and Fuzzy PID control simulation were established in MATLAB/Simulink. Longitudinal steering simulation was carried out on a slope (0°, 30°), and an analysis of the simulation of lateral steering along the contour line was carried out. The simulation results showed that this steering control strategy was able to automatically adjust the target vehicle speed to avoid rollover while the driver was inputting steering signals.

**Keywords:** tracked vehicles; hydrostatic drive; differential steering; control strategy



## 1. Introduction

Tracked vehicles have good all-terrain and off-road capabilities and are widely used in civilian and military applications [1]. Differential steering for tracked vehicles is a common steering method for tracked vehicles which enables the vehicle to be steered while in motion. The pump-motor hydrostatic speed control circuit is an important method of travel, providing differential steering for tracked vehicles [2,3]. Due to the complex terrain over which tracked vehicles travel, the theoretical analysis and control of tracked vehicles steering has become a research issue in recent years. Steering control has been thoroughly studied for tracked vehicles. Zhai L [4] designed a steering coupling-based direct average torque distribution control strategy that can significantly improve the lateral stability of dual-motor-driven high-speed electric tracked vehicles. Chen Z Y proposed a steering control strategy based on the BP neural network modified with the PID algorithm for a dual electric tracked vehicle [5]. Morimoto T et al. [6] proposed a method to intervene in the input reference based on vehicle attitude information to prevent tracked vehicles from skidding while steering on slopes. Park W.Y. et al. [7] theoretically analyzed tracked vehicle steering over different road surfaces to obtain steady-state steering characteristics. Li Guoqiang [8] established a mechanical model for high-speed steering of tracked vehicles and studied the relationship between the steering radius, steering angular velocity and critical velocity at which drifting occurs. Liu Yi et al. [9] proposed a control strategy that adjusted the motor drive torque to regulate the angular velocity of the steering for tracked vehicles based on ground steering resistance. Gai Jiangtao et al. [10] proposed a control strategy that improved the control accuracy of automatic driving trajectory

tracking by considering the slip–slide of track differential steering. When researching the steering safety of tracked vehicles, Su Yong et al. [11] studied the mechanical process of high-speed steering of tracked vehicles on slopes using multi-body dynamic modeling and simulation, and analyzed the relationship between vehicle speed, steering radius and centrifugal force, track tension, and the slope gradient of ground. Sun Fengchun et al. [12] studied the interrelationship between the center of gravity position, vehicle speed, steering radius, and vehicle posture on the slope, and analyzed steering instability factors. Zhang Yu et al. [13] analyzed the minimum instantaneous radius law for avoiding slip under different ground and slope conditions using the drive force–slip rate equation of the vehicle ground mechanics. Zeng Gen et al. [14] proposed a prediction method for the dual side drive force of tracked vehicles when steering on slopes that could be used to control the drive force of the motor.

In summary, studies have focused on the analysis of the mechanism of steering in tracked vehicles over different terrains, including factors such as track skidding, slipping, steering radius, and driving force, and have emphasized the response characteristics and smoothness of the steering adjustment process. In addition, the factors influencing the safety and instability of steering on slopes have been analyzed. However, less research has been devoted to control strategies considering the safety of steering skid and rollover of tracked vehicles on complex terrain. Therefore, this paper proposes a differential steering control strategy that avoids sideslip and rollover of the tracked vehicles on sloped terrain, while integrating the driver's steering intentions. Section 2 presents materials and methods; Section 3 presents the modeling and results analysis; Section 4 presents an analysis of the control strategy on the basis of steering stability; Section 5 presents a discussion; and Section 6 draws the conclusions.

## 2. Materials and Methods

### 2.1. Operating Principle and Modeling of the Dual-Sided Independent Drive System of a Tracked Vehicle

Dual-sided independent drive steering of tracked vehicles is achieved by adjusting the speed of the drive wheels on both sides. The structure of the dual variable pump–quantitative motor hydrostatic drive system for tracked vehicles studied in this paper is shown in Figure 1.

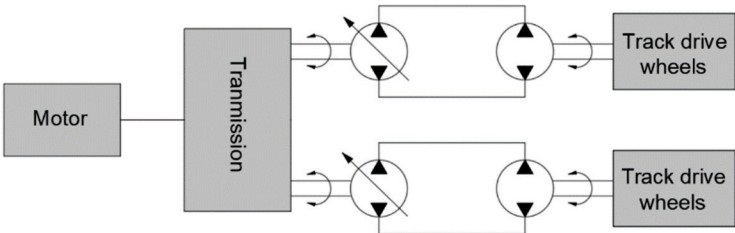

**Figure 1.** The driving system of a tracked vehicle.

As shown in Figure 1, the engine transmits the power to the transmission, and the output shaft of the transmission is connected in parallel with two bi-directional variable pumps. The variable pumps and motor form a hydrostatic closed circuit, which drives the speed and steering of the hydraulic motors through the pumps, and the hydraulic motors are directly connected with the track drive wheels to drive the vehicle. For the subsequent theoretical study of the whole vehicle steering control, a mathematical model of the closed-loop system of pump-controlled motors on the tracked vehicle needs to be analyzed.

Variable pump flow model:

$$q_p = x D_p \omega_p - C_{ip}(P_h - P_s) - C_{ep}P_h \tag{1}$$

Motor working flow model:

$$q_m = q_p = C_{im}(P_h - P_s) + C_{em}P_h + D_m\frac{d\theta_m}{dt} + \frac{V_0}{\beta_e}\frac{dP_h}{dt} \tag{2}$$

Motor and load torque balance equation:

$$D_m(P_h - P_s) = J_m\frac{d^2\theta_m}{dt^2} + B_m\frac{d\theta_m}{dt} + T_l$$

In the above Equations (1)–(3), $D_p$ is the variable pump displacement; $x$ is the variable pump displacement adjustment factor; $D_m$ is the motor displacement; $\omega_p$ is the pump input speed; $\theta_m$ is the motor rotation angle; $C_{ip}$ is the pump internal leakage coefficient; $C_{ep}$ is the pump external leakage coefficient; $C_{im}$ is the motor internal leakage coefficient; $C_{em}$ is the motor external leakage coefficient; $P_h$ and $P_s$ are pressure values at high and low sides respectively; $V_o$ is the total volume of high pressure side; $\beta_e$ is the hydraulic oil bulk modulus of elasticity; $J_m$ is the total rotational inertia of motor and load; $B_m$ is the total viscous damping factor of motor and load; $T_l$ is the external load torque acting on the motor shaft.

### 2.2. Differential Steering Model of a Tracked Vehicle
Kinematic Analysis of Differential Steering

During the differential steering of a tracked vehicle, the outer track skids and the inner track slips; the steering kinematics of the tracked vehicle are shown in Figure 2.

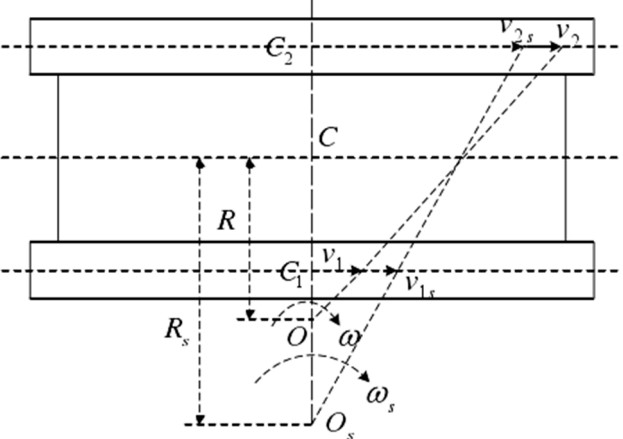

**Figure 2.** Kinematic analysis of tracked vehicle steering.

In Figure 2, $V_1$ and $V_2$ are the theoretical traveling speeds of the inner and outer tracks, respectively; $O$ is the theoretical steering center of the vehicle; $R$ is the theoretical steering radius; $\omega$ is the theoretical angular velocity of the tracked vehicle steering; $V_{1s}$ and $V_{2s}$ are the actual traveling speeds of the inner and outer tracks, respectively; $O_s$ is the actual steering center of the vehicle; $R_s$ is the actual steering radius; $\omega_s$ is the actual steering angular velocity of the tracked vehicle.

According to Bwkker theory [15], The track slip rate when the vehicle is in motion can be calculated using the following equation:

$$i = -\frac{K/L}{\ln(F/F_{\max})} \tag{3}$$

In the formula, $K$ is the ground shear deformation coefficient; $L$ is the track grounding length; $F$ is the traction.

Supposing $\omega_1$ and $\omega_2$ are the rotation angular velocities of the inner and outer driving wheels; $r$ is the radius of the driving wheel, so the actual speeds of the inner and outer tracks can be expressed as:

$$\begin{cases} v_{1s} = r\omega_1(1+i_1) \\ v_{2s} = r\omega_2(1-i_2) \end{cases} \tag{4}$$

If the track gauge on both sides is $B$, the actual turning radius is:

$$R_s = \frac{B}{2} \cdot \frac{v_{2s} + v_{1s}}{v_{2s} - v_{1s}} \tag{5}$$

The tangential velocity of the mass center along the steering trajectory is:

$$v_{cs} = \frac{v_{1s} + v_{2s}}{2} \tag{6}$$

The steering angular velocity is:

$$\omega = \frac{v_{2s} - v_{1s}}{B} \tag{7}$$

### 2.3. Dynamics Analysis of Slope Steering

The steering of tracked vehicles on slopes can be divided into four typical conditions, as shown in Figure 3a. The steering process of the tracked vehicle is influenced by the slope of the ground, the driving force, the drag force, and the centrifugal force. A schematic diagram of the spatial steering of tracked vehicles is shown in Figure 3b, and a sketch of the steering dynamics analysis is presented in Figure 3c.

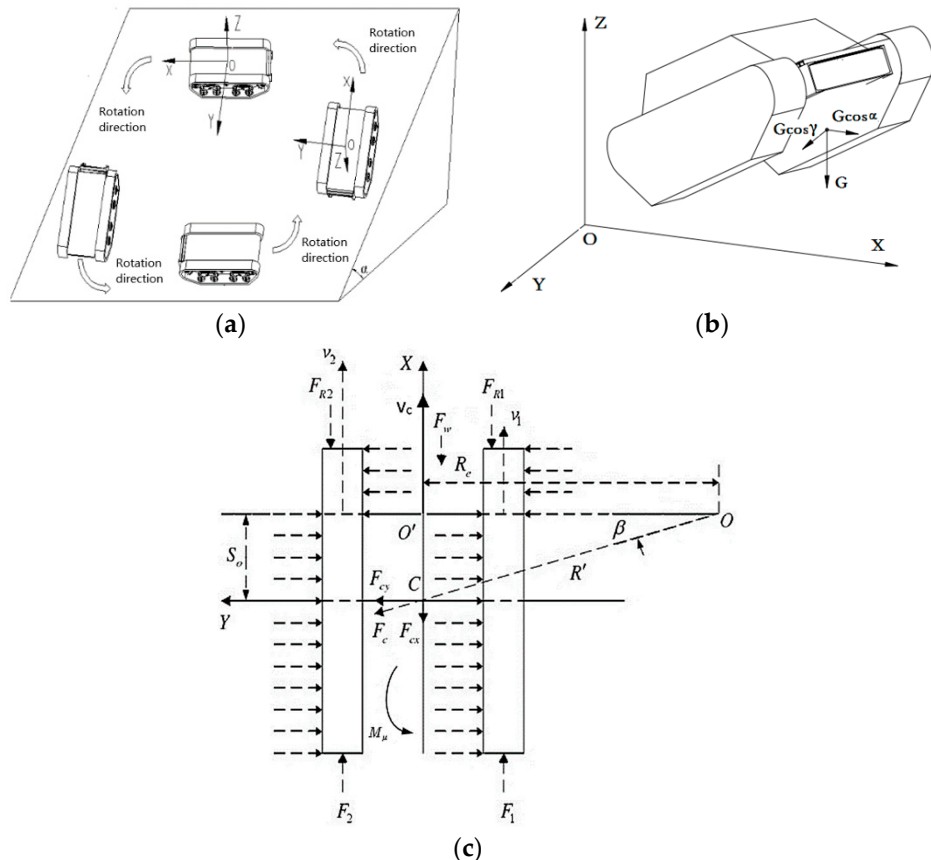

**Figure 3.** (**a**) Typical steering conditions of tracked vehicles on a slope; (**b**) gravity and component directions of tracked vehicles; (**c**) ground steering force diagram of tracked vehicles.

In Figure 3b, $O - XYZ$ is the attitude coordinate system based on the tracked vehicle body; let the angle of the tracked vehicle $OY$ axis relative to the horizontal plane be $\gamma$, the angle of the ox axis relative to the horizontal plane be $\alpha$; $G$ is the weight of the whole vehicle, and the component forces of $G$ on the $OX$, $OY$ axis are Gcos$\alpha$ and Gcos$\gamma$. In Figure 3c, $F_1$ and $F_2$ are the driving forces of the tracks on both sides; $F_{R1}$ and $F_{R2}$ are the rolling resistance of the tracks; $M_\mu$ is the steering resistance moment; $R_e$ is the distance from the steering center to the vehicle centerline ($X$-axis in the figure). $R'$ is the distance from the center of the vehicle plane (point C) to the steering center (O); $S_O$ is the longitudinal offset of the steering center; $\beta$ is the angle between the direction of the vehicle centrifugal force and the $Y$-axis; $h$ is the height of the mass center. The expressions for the driving force and rolling resistance on both sides are:

$$F_1 = \begin{cases} 0.5\varphi G \cdot \cos\gamma \quad \frac{T_1 i\eta}{r} > 0.5\varphi G \cdot \cos\gamma \\ \frac{T_1 i\eta}{r} \quad \frac{T_1 i\eta}{r} \leqslant 0.5\varphi G \cdot \cos\gamma \end{cases} \tag{8}$$

$$F_2 = \begin{cases} 0.5\varphi G \cdot \cos\gamma \quad \frac{T_2 i\eta}{r} > 0.5\varphi G \cdot \cos\gamma \\ \frac{T_2 i\eta}{r} \quad \frac{T_2 i\eta}{r} \leqslant 0.5\varphi G \cdot \cos\gamma \end{cases} \tag{9}$$

$$F_{R1} = F_{R2} = \frac{fG \cos\alpha}{2} \tag{10}$$

In Equations (9)–(11), $T_1$ and $T_2$ are motor torque on both sides, respectively; $\varphi$ is ground adhesion coefficient; $i$ is the side transmission ratio; $r$ is the radius of the drive wheel; $\eta$ is the transmission efficiency from motor to track, where $\eta = 0.941(0.97 - 0.03v)$ [16]; $f$ is the rolling resistance coefficient.

The longitudinal and transverse components of the centrifugal force are:

$$\begin{cases} F_{cx} = mR'\omega^2 \sin\beta \\ F_{cy} = mR'\omega^2 \cos\beta \end{cases} \tag{11}$$

The longitudinal offset of the steering center is:

$$S_o = \frac{LF_{cy}}{2\psi G \cdot \cos\alpha} \tag{12}$$

In Equation (13), $L$ is the grounding length of the tracks; $\psi$ is the lateral attachment coefficient; $G$ is the fully loaded weight of the vehicle.

According to the literature [17], the steering resistance moment can be expressed as:

$$M_\mu = \frac{1}{4}\mu G \cdot \cos\alpha \cdot L \left[ 1 - \left( \frac{2S_o}{L} \right)^2 \right] \tag{13}$$

The steering resistance coefficient can be calculated from the Nifty empirical formula [15]:

$$\mu = \frac{\mu_{\max}}{0.925 + 0.15\rho} (\rho \geqslant 0.5) \tag{14}$$

In Equation (15), $\rho$ is the relative steering radius, $\rho = \frac{R}{B}$.

From the dynamic equilibrium equation of the tracked vehicle, the driving force of the both sides tracks can be expressed as:

$$\begin{cases} F_2 + F_1 - (F_{R1} + F_{R2} + F_w \pm G \cdot \sin\alpha) - F_{cx} = m\dot{v}_c \\ (F_2 - F_1)\frac{B}{2} + (F_{R1} - F_{R2})\frac{B}{2} - M_\mu = J\dot{\omega} \end{cases} \tag{15}$$

## 3. Modeling and Results Analysis

### 3.1. Analysis of the Control System

The input from the driver's driving and steering signals can be interpreted as the desired speed input for the motors on both sides, and the variable pump controller sends a responsive displacement command to the variable pump mechanism, thereby regulating the speed of the motors in order to accomplish the intentions of the driver. The closed-loop control system of vehicle steering and driving is shown in Figure 4.

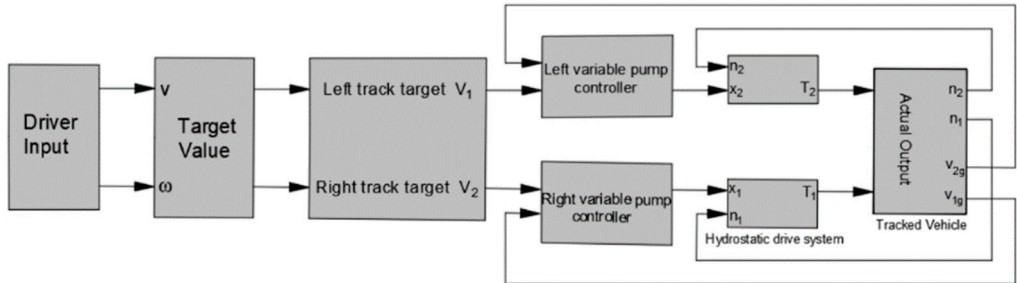

**Figure 4.** Control system diagram of tracked vehicle steering.

As shown in Figure 4, the intention of the driver can be converted into the desired vehicle speed $v$ and the desired steering angular velocity $\omega$, which are decomposed into the desired track speeds on both sides of the tracked vehicle by means of the differential speed algorithm, and the actual speed of the driving wheels can be adjusted by means of feedback control in the driving wheels so as to achieve the required speed and steering control.

### 3.2. Simulation Model Building

On the basis of an analysis of the whole vehicle steering driving closed-loop control system, the mathematical models of the hydrostatic drive system, the steering kinematics, and the dynamics of the tracked vehicle in the previous section are used as sub-model modules in the tracked vehicle steering driving system, and the simulation model is built in MATLAB/Simulink, as shown in Figure 5.

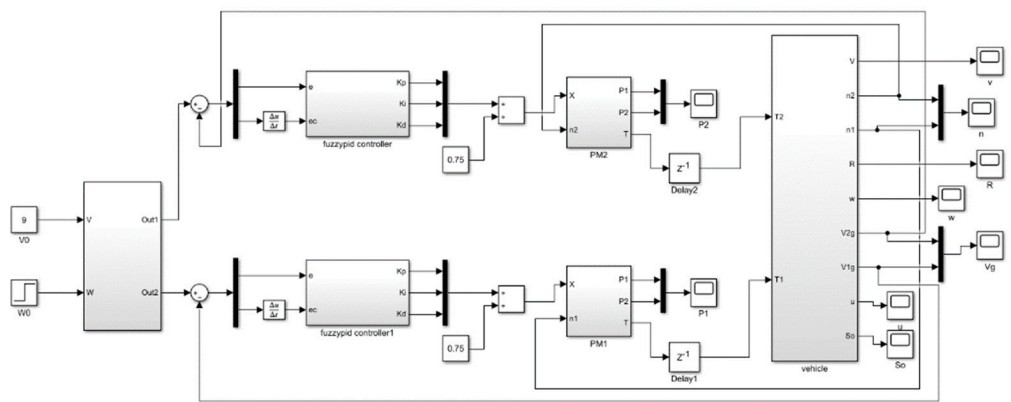

**Figure 5.** Simulation model of vehicle steering driving system.

On the basis of Figure 5, it can be seen that the vehicle steering driving system simulation model includes a control strategy input module ($v - w$), a pump displacement controller module (Fuzzy-PID controller), a hydrostatic drive system module (PM), and a vehicle steering kinematics and dynamics module (vehicle). The control strategy module interprets the steering input in the form of the appropriate track speed required for both sides while avoiding vehicle rollover, and sends the corresponding variable pump displacement control signal to the variable pump controller; then, the signal from the motor speed sensor is fed to the controller to complete the closed-loop control of vehicle speed

and steering conditions. The main parameters of the tracked vehicle used in the simulation model are shown in Table 1, below.

**Table 1.** Main parameters of the tracked vehicle.

| Parameters | Value | Unit |
|---|---|---|
| Tracked vehicle overall mass $G$ | 2800 | kg |
| High quality core $h$ | 1.15 | m |
| Track center distance $B$ | 1.25 | m |
| Crawler ground length $L$ | 1.56 | m |
| Drive wheel radius $r$ | 0.18 | m |
| Variable Pump Displacement | 0–71 | mL/r |
| Hydraulic system pressure | 30 | Mpa |
| Motor Displacement | 514 | mL/r |
| Maximum pressure of fixed motor | 31.5 | Mpa |

## 4. Analysis of Control Strategy Based on Steering Stability

The steering safety performance of tracked vehicles refers to the ability of vehicle steering to avoid possible skidding or rollover under the influence of centrifugal force and terrain slope factors, which can be avoided by formulating a suitable steering control strategy. During the control process, the real-time conditions of vehicle steering are analyzed to determine dangerous operation conditions such as skidding and rollover, and the instructions for avoiding skidding and rollover are transmitted to the driving system controller.

### 4.1. Analysis of Skid Conditions

The analysis of Figure 3 and Equations (12) and (14) show that skidding of the tracked vehicle when steering needs to consider the longitudinal shift of the instantaneous steering center caused by the lateral force of centrifugal force, and when its value reaches the maximum, the lateral force $F_{cy}$ generated by centrifugal force is equal to the maximum lateral adhesion force provided by the ground. The steering driving condition of the tracked vehicle is in the critical state of side slipping, and the center speed of the tracked vehicle at this time is referred to as the critical speed of skidding steering. Therefore, the conditions for the steering of a tracked vehicle without skidding [18] are:

$$S_o = \frac{LF_{cy}}{2\psi G \cdot \cos \alpha} \leqslant \frac{L}{2} \tag{16}$$

### 4.2. Analysis of the Rollover Condition

On the basis of the analysis of Figure 3 and Equations (13) and (14), it can be seen that the occurrence of rollover needs to consider the changes in the normal load of the inner and outer track joints caused by the lateral component of centrifugal force and the lateral component of gravity, and when the rollover moment of a tracked vehicle while turning is equal to the return moment provided by gravity, and the normal load of the inner track joints is zero, the tracked vehicle is in the critical state of being in danger of rollover, and the center speed of the tracked vehicle at this time is referred to as the critical speed of lateral tilting steering; then, the condition for turning a tracked vehicle without rollover occurring [19] is:

$$\frac{F_{cy}h \pm G \sin \alpha}{B} \leqslant \frac{G \cdot \cos \alpha}{2} \tag{17}$$

### 4.3. Development of an Integrated Control Strategy Considering Steering Safety

According to the operating environment of the tracked vehicle, the road conditions are: $f = 0.1$; $\mu_{max} = 0.8$; $\psi = 0.9$ and when $\alpha = 0°$, the critical speed-steering radius and steering angular velocity relationships under the influence of skidding and rolling are as shown in Figures 6 and 7.

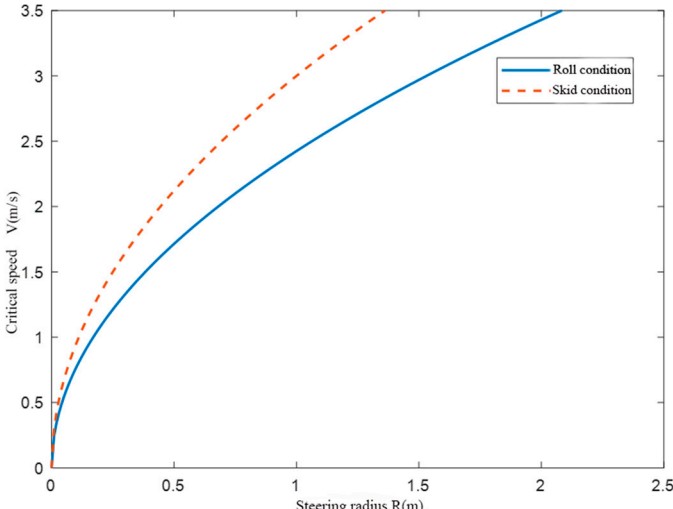

**Figure 6.** $v - R$ target curve under skidding conditions.

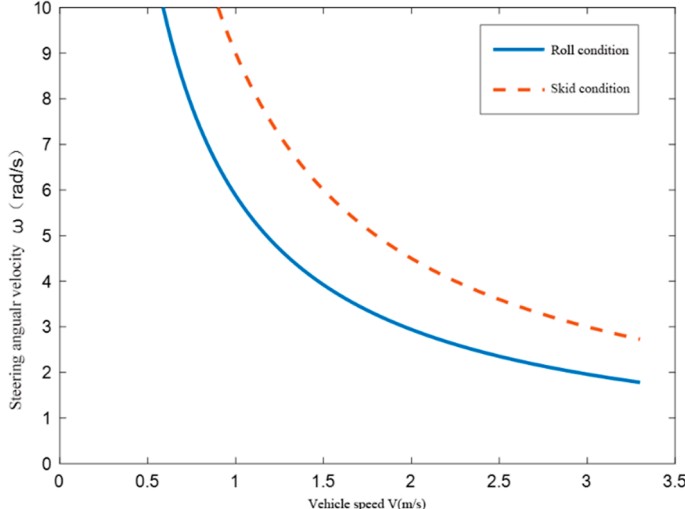

**Figure 7.** $v - \omega$ target curve under rollover conditions.

On the basis of Figures 6 and 7, at the same steering angular velocity, it can be seen that the critical speed under the rollover condition is always lower than the critical speed under the skidding condition. Therefore, only the relationship between critical speed and steering angular velocity under the rollover condition needs to be considered. When $\alpha \neq 0°$, the research objective is the steering control of the tracked vehicle under travelling conditions that are longitudinal along the slope and along the contour line.

On the basis of the analysis of the rollover conditions mentioned above, and considering the steering rollover control strategy of a tracked vehicle under different slopes, the control surfaces are calculated and fitted, as shown in Figure 8, for steering a tracked vehicle longitudinally along the slope and steering up and down the slope, respectively, along the contour line under different slope degrees ($\alpha = 0 \sim 30°$). The slope degree $\alpha$ can be detected using an angle sensor in the actual situation. Figure 8a presents the control surface for rollover condition under longitudinal travel along the slope; Figure 8b presents the control surface for the rollover condition under travel along the contour line turning up the slope; Figure 8c presents the control surface for the rollover condition under travel along the contour line turning down the slope.

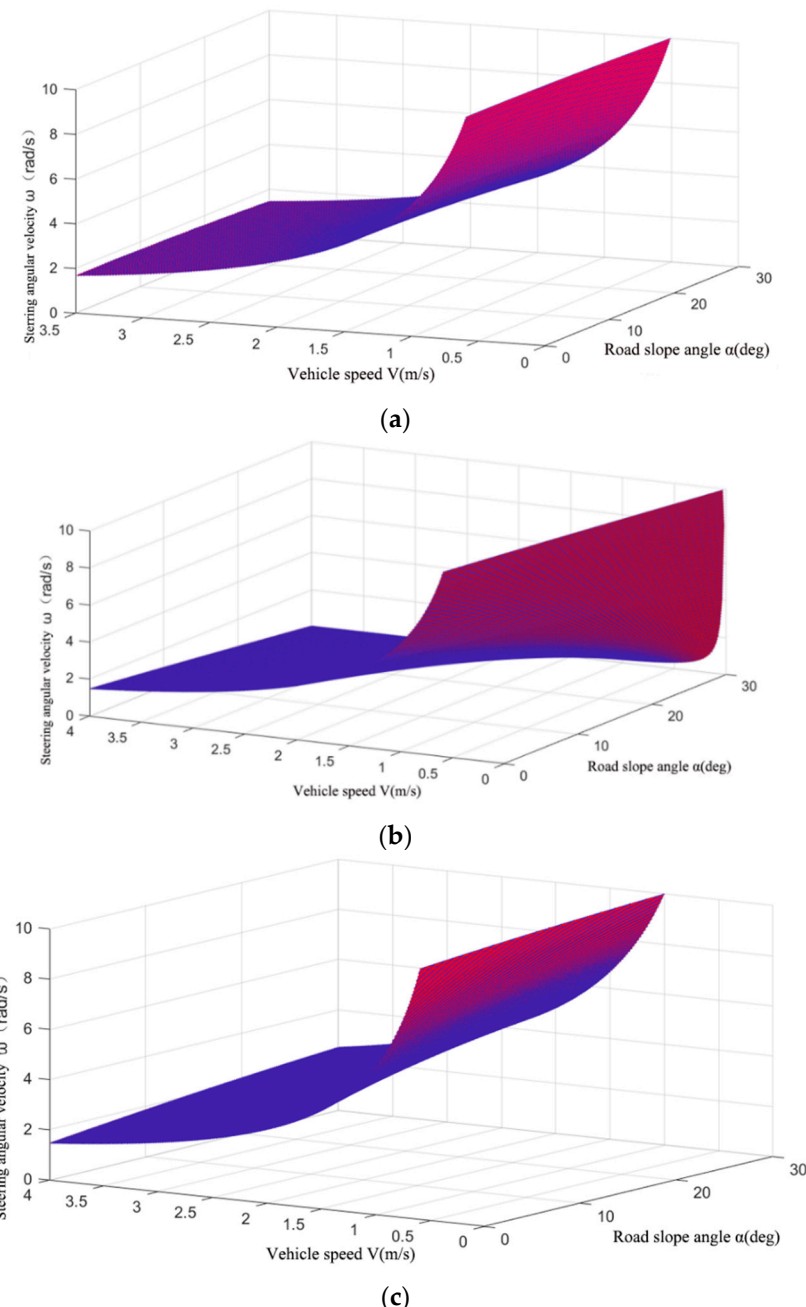

**Figure 8.** Slope steering driving roll control surface: (**a**) steering roll control surface for longitudinal driving along the slope; (**b**) roll control surface for turning uphill along the contour line; (**c**) roll control surface for steering downhill along the contour line.

According to the fitting relationship in Figure 8, the control strategy rule for the safety condition preventing rollover during steering is obtained as follows:

$$
\omega = 
\begin{cases}
1 \, (v \leqslant 0.6) \\
\begin{cases}
v + 0.4 \left( 0.6 < v \leqslant \dfrac{-0.4 + \sqrt{0.16 + 23.52 \cos \alpha}}{2} \right) \\
\dfrac{5.88 \cos \alpha}{v} \left( \dfrac{-0.4 + \sqrt{0.16 + 23.52 \cos \alpha}}{2} < v \leqslant 3.3 \right)
\end{cases} \\
\begin{cases}
v + 0.4 \left( 0.6 < v \leqslant \dfrac{-0.4 + \sqrt{0.16 + (23.52 \cos \alpha \pm 39.2 \sin \alpha)}}{2} \right) \\
\dfrac{5.88 \cos \alpha \pm 9.8 \sin \alpha}{v} \left( \dfrac{-0.4 + \sqrt{0.16 + (23.52 \cos \alpha \pm 39.2 \sin \alpha)}}{2} < v \leqslant 3.3 \right)
\end{cases}
\end{cases}
\tag{18}
$$

On the basis of the above analysis, the control rules are written using the S-function in MATLAB, corresponding to the $v - w$ module in Figure 5, the control objectives of which are vehicle speed and steering angular velocity.

## 5. Discussion

To verify the effectiveness of the control strategy, the vehicle steering performance was simulated and the road parameters were selected as shown in Table 2.

**Table 2.** Road parameters in simulation.

| Parameters | Value |
|---|---|
| Rolling resistance coefficient $f$ | 0.1 |
| Steering resistance coefficient $\mu_{max}$ | 0.8 |
| Lateral adhesion coefficient $\psi$ | 0.9 |

The simulation and verification of the control strategy was performed under four typical steering conditions with different slope angles on the ground, as follows:

(1)  The simulation conditions are: the initial vehicle speed is 3 m/s, when t = 10 s, the steering angle speed input by the driver is 2.5 rad/s. If the angle sensor detects $\alpha = 0°$, According to Formula (19), the maximum steering angular speed corresponding to the vehicle speed of 3 m/s is 1.96 rad/s, which does not achieve the target steering angular speed of 2.5 rad/s. Therefore, to achieve the desired steering command, the vehicle speed needs to be adjusted automatically. Figure 9a shows the curve of the steering control signal, the inside and outside speed of the track, and the central vehicle speed under this condition. Figure 9b shows the pressure change curve of the hydraulic motors on both sides during steering.

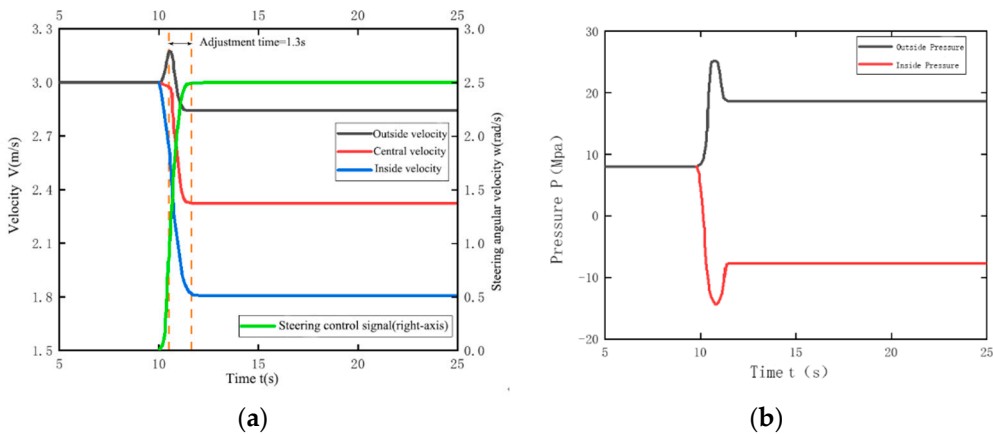

(a)  (b)

**Figure 9.** Simulation curves of operation under Condition 1: (**a**) simulation curve of vehicle speed and steering angular velocity; (**b**) simulation curve of pressure on both sides.

The simulation results in Figure 9a show that the steering angular speed input by the driver increases from 0 to 2.5 rad/s in the interval 10–10.5 s, and because the control signal had not reached 1.96 rad/s, the control strategy does not work in the interval between 10 and 10.5 s; the inner speed is decreased and the outer speed is increased, while the vehicle speed remains unchanged during steering, which fits the characteristics of differential steering. When the angular velocity exceeds 1.96 rad/s, the control rules automatically adjust the driving speed so that the speeds of the inner and outer tracks change simultaneously, while the vehicle speed is automatically adjusted to 2.35 m/s in 1.3 s, completing the desired steering requirement of 2.5 rad/s. At the same time. Figure 9b shows that the pressures of the hydraulic motors on both sides do not exceed the pressure limit of 31.5 Mpa.

(2) The simulation conditions are: Initial speed is 2.8 m/s, when t = 10 s, the steering angle speed input by the driver is 2.2 rad/s. If the angle sensor detects $\alpha = 30°$ for uphill driving steering, according to Formula (19), the maximum steering angular speed corresponding to the vehicle speed of 2.8 m/s is 1.82 rad/s, which does not achieve the target steering angular speed of 2.2 rad/s. Therefore, to perform the desired steering command, the vehicle speed needs to be adjusted automatically. Figure 10a is the curve of the steering control signal, the inside and outside speeds of the track, and the central vehicle speed in this condition. Figure 10b shows the pressure change curve of the hydraulic motors on both sides during steering.

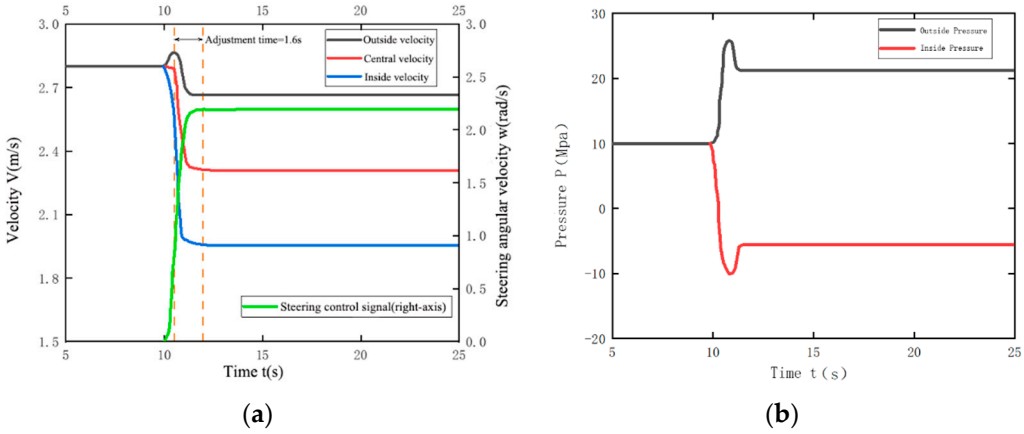

**(a)**                                        **(b)**

**Figure 10.** Simulation curves of operation under Condition 2: (**a**) simulation curve of vehicle speed and steering angular velocity; (**b**) simulation curve of pressure on both sides.

On the basis of the simulation results presented in Figure 10a, the steering angular speed inputs by the driver increase from 0 to 2.2 rad/s in the interval 10–10.5 s. Because the steering control signal input has not yet reached 1.82 rad/s, the control strategy does not work in the interval between 10 and 10.5 s; the inside speed is decreased and the outside speed is increased, while the vehicle speed remains unchanged during steering, which fits the characteristics of differential steering. When the angular velocity exceeds 1.82 rad/s, the speed of the vehicle is automatically adjusted to 2.31 m/s over 1.6 s, because the control rule automatically adjusts the driving speed so that the speeds of the inner and outer tracks change at the same time, and a steering angular velocity of 2.2 rad/s is achieved by reducing the vehicle speed. In addition, Figure 10b shows that the pressures of the hydraulic motors on both sides increase and then decrease, and do not exceed the pressure limit of 31.5 MPa.

(3) The simulation conditions are: Initial vehicle speed is 1m/s, when t = 10 s, driver inputs a steering control signal of 0.5 rad/s. If the angle sensor detects $\alpha = 30°$, and when driving along the slope contour and turning upward, according to Formula (19), the maximum steering angular speed corresponding to the vehicle speed of 1m/s is 0.2 rad/s, which does not achieve the target steering angular speed of 0.5 rad/s. To perform the desired steering command, the vehicle speed needs to be adjusted automatically.

The simulation results in Figure 11a show that the steering angular speed input by the driver increases from 0 to 0.5 rad/s within the interval 10–10.5 s. Because the driver input has not yet reached 0.2 rad/s, the control strategy does not work in the interval between 10 and 10.5 s; the inner speed is decreased and the outer speed is increased, while keeping the vehicle speed constant during steering, which fits the characteristics of differential steering. In addition, when the angular velocity exceeds 0.2 rad/s, the control rule automatically adjusts the driving speed so that the speeds of the inner and outer tracks change at the same time, so that the vehicle speed is automatically adjusted to 0.38 m/s within 2 s, and the steering target of 0.5 rad/s for the whole vehicle is achieved by reducing the driving

speed. Figure 11b shows that the pressure of the hydraulic motors on both sides increases and then decreases without exceeding the pressure limit of 31.5 Mpa.

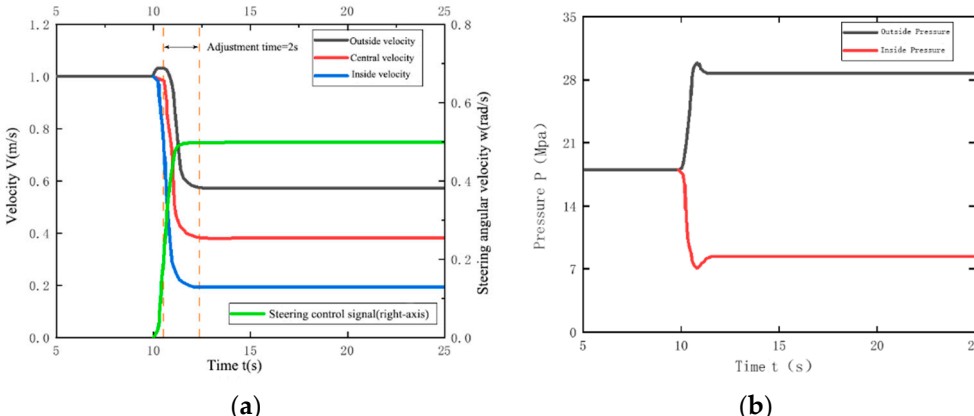

**Figure 11.** Simulation curves of operation under Condition 3; (**a**) simulation curve of vehicle speed and steering angular velocity; (**b**) simulation curve of pressure on both sides.

(4) The simulation conditions were: At an initial speed of 3.3 m/s and t = 10 s, the driver inputs a steering angular velocity control signal as the steering intention, the value of which increases from 0 to 3.2 rad/s within 0.5 s. If the angle sensor detects $\alpha = 30°$ for steering downhill along the contour line, according to Formula (19), the maximum steering angular speed corresponding to the vehicle speed of 3.3 m/s is 3 rad/s, which does not achieve the target steering angular speed of 3.2 rad/s. To achieve the target steering, the vehicle speed needs to be adjusted automatically.

The simulation results in Figure 12a show that when the driver inputs the steering control signal, the signal has not reached 3 rad/s, and therefore the control strategy is not triggered, in the interval between 10 and 10.5 s. To keep the vehicle speed constant during steering, the inner speed is decreased and the outer speed is increased, which is in line with the characteristics of differential steering. When the input angular velocity exceeds 3 rad/s, the control rule automatically adjusts the driving speed so that the inner and outer track speeds change at the same time, so that the vehicle center speed is automatically adjusted from 3.3 m/s to 3.1 m/s within 1.5 s. By reducing the vehicle speed, the steering target requirement of 3.2 rad/s is achieved, which shows that the control strategy is effective. Figure 12b shows that the pressure of the hydraulic motors on both sides increases and then decreases without exceeding the pressure limit of 31.5 Mpa.

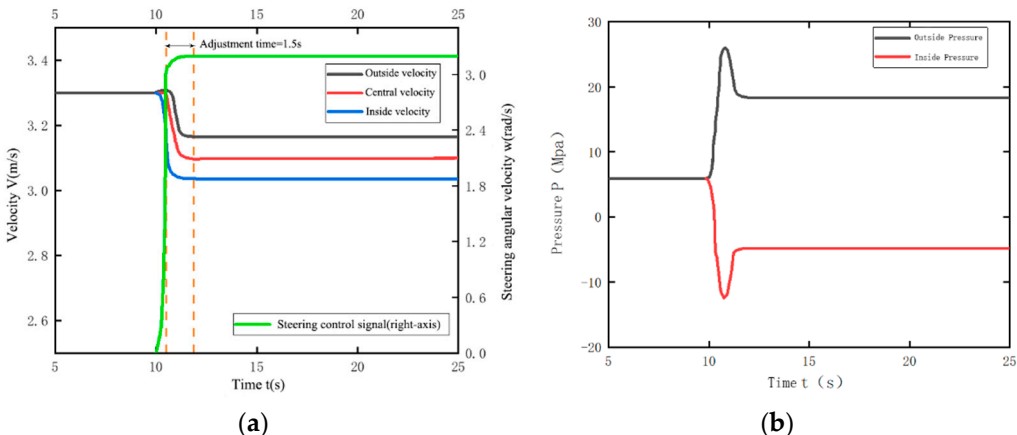

**Figure 12.** Simulation curves of operation under Condition 4: (**a**) simulation curve of vehicle speed and steering angular velocity; (**b**) simulation curve of pressure on both sides.

## 6. Conclusions

A control strategy considering steering safety on sloped terrain is proposed for dual independently driven tracked vehicles. The vehicle speed and steering angular velocity of the tracked vehicle were considered under different constraints such as slope and travelling speed, so as to establish a safe steering control strategy for driving on sloped terrain. Simulation verification of the safe steering control strategy was carried out on slopes of $0°$ and $30°$ for longitudinal driving steering along the slop and driving up and down the slope along the contour line, respectively. The simulation results show that when the control signal of the driver exceeds the vehicle's speed constraint with respect to rollover during steering, which is automatically adjusted according to the control rule, the steering intention of the driver can be satisfied by reducing the vehicle speed within a certain adjustment stability time, thus verifying the feasibility of the proposed safe steering control strategy for tracked vehicles in different terrains.

**Author Contributions:** J.H.: Conceptualization, Supervision, Methodology. F.W.: Software, Writing—review and editing, Validation, Data curation. Y.W.: Formal analysis, Software, Writing—original draft. All authors have read and agreed to the published version of the manuscript.

**Funding:** This research was funded by the foundation of Science and Technology Program of Jiangsu Province, grant number: SZ-YC202165.

**Institutional Review Board Statement:** Not applicable.

**Informed Consent Statement:** Not applicable.

**Acknowledgments:** The authors would like to thank Xia Changgao at Jiangsu University for their valuable suggestions and help. The work was supported by foundation of Science and Technology Program of Jiangsu Province (No:SZ-YC202165).

**Conflicts of Interest:** The authors declare no conflict of interest.

## Nomenclature

| Symbol | Meaning |
|---|---|
| $D_p$ | Variable pump displacement |
| $D_m$ | Motor displacement |
| $\omega_p$ | Pump input speed |
| $\theta_m$ | Motor rotation angle |
| $C_{ip}$ | Pump internal leakage coefficient |
| $C_{ep}$ | Pump external leakage coefficient |
| $C_{im}$ | Motor internal leakage coefficient |
| $C_{em}$ | Motor external leakage coefficient |
| $V_0$ | Total volume of high-pressure side |
| $\beta_e$ | Hydraulic oil bulk modulus of elasticity |
| $J_m$ | Total rotational inertia of motor and load |
| $B_m$ | Total viscous damping factor of motor and load |
| $T_l$ | External load torque acting on the motor shaft |
| $\omega_{1,2}$ | Rotation angular velocities of the inner and outer driving wheels |
| $v_{1,2}$ | Theoretical speed of inner and outer tracks relative to the ground |
| $r$ | Driving wheel radius |
| $v_{1s,2s}$ | Actual speed of inner and outer tracks relative to the ground |
| $R_s$ | Actual turning radius |
| $\omega_s$ | Actual steering angular velocity of tracked vehicle |
| $B$ | Track gauge of both sides |
| $F_{1,2}$ | Driving forces of the tracks on both sides |
| $F_{R1, R2}$ | Rolling resistance of the tracks |
| $M_\mu$ | Steering resistance moment |
| $R_e$ | Distance from the steering center to the vehicle centerline |
| $R'$ | Distance from the center of the vehicle plane |

| $S_o$ | Longitudinal offset of the steering center |
|---|---|
| $K$ | Ground shear deformation coefficient |
| $\beta$ | Vehicle deflection angle |
| $G$ | Vehicle weight |
| $\gamma$ | Angle with Y-axis |
| $\alpha$ | Angle with X-axis |
| $h$ | Height of the mass center |
| $T_{1,2}$ | Motor torque on both sides |
| $\varphi$ | Ground adhesion coefficient |
| $i$ | Side transmission ratio |
| $\eta$ | Transmission efficiency from motor to track |
| $f$ | Rolling resistance coefficient |
| $L$ | Track grounding length |
| $\rho$ | Relative steering radius |
| $\psi$ | Lateral adhesion coefficient |

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
