# Peer review of "A Control Method for the Differential Steering of Tracked Vehicles Driven Independently by a Dual Hydraulic Motor"

_applsci, doi:10.3390/app12136355_

Round 1
Reviewer 1 Report
Please check all images for legibility and expression. You can also rewrite the discussion section. Two sentences should not begin with "In this paper". Apart from that, I think that your work can provide benefits to those who work with the subject.
Author Response
Thanks for your comments. I've checked all the images and their expressions. According to your suggestions, the discussion and conclusion section have been modified. See the marking yellow section in the article for specific modifications. The last part of the manuscript has also been modified. See the marking yellow section for the specific modified parts.

Reviewer 2 Report
In the article under review, the authors present the results of research aimed at solving the problem of improving the steering control strategy of a tracked vehicle independent driven by a hydraulic motor in order to avoid its rollover. The studies were carried out by mathematical modeling using the MATLAB/Simulink software.
The results of the presented studies can be useful to a wide range of readers - scientists and practitioners, specialists in the field of vehicle drives control systems.
In the Introduction and literature review, the prerequisites for conducting research are considered in sufficient detail, and the purpose of the paper is formulated. The main parts of the paper provide a description of the driving system of a tracked vehicle. The steering kinematic analysis is given, as well as a dynamic analysis of slope steering is given too. An integrated control strategy has been proposed to ensure the safe movement of a tracked vehicle on a slope. A mathematical model has been developed and simulation results for various driving conditions have been presented.
During the review, I was left with the questions and I drew attention to the following shortcomings, the correction of which would improve the quality of the article:
- The paper considers the case when the ground adhesion coefficient Ï• is the same for both sides. Is it necessary to make adjustments to the control strategy if this coefficient is different? If this is the case, then appropriate explanations should be made in the paper.
- In the control strategy (19) recommended by the authors, different values of ω are set for the same value of the velocity v. This needs some explanation.
- It is required to substantiate the choice of PID controller by the authors. Why was it not enough to use a PI or P controller?
- The authors did not present the results of experimental studies confirming the correctness of theoretical studies.
Author Response
Thank you very much for your comments and taking the time to read our article.
Comment 1:
The paper considers the case when the ground adhesion coefficient Ï• is the same for both sides. Is it necessary to make adjustments to the control strategy if this coefficient is different? If this is the case, then appropriate explanations should be made in the paper.
Answer: Thanks for your comments. The answer to this question is as follows. This paper studies the safe driving control strategy to prevent rollover during steering, which mainly depends on the relationship between the rollover torque and the aligning torque of the tracked vehicle, that is, whether the normal load of the inner track is greater than zero. Rollover is related to the lateral component of the centrifugal force of the tracked vehicle and the slope angle. The adhesion coefficient affects the ground driving force of the tracks on both sides. The slip rate of the tracks on both sides has been considered in the process of building the vehicle driving system model. Therefore, the ground adhesion coefficient on both sides does not directly affect the analysis and formulation of this control strategy.
Comment 2: In the control strategy (19) recommended by the authors, different values of ω are set for the same value of the velocity v. This needs some explanation.
Answer: Thanks for your valuable comments. The answer to this question is as follows: In this paper, a control strategy to prevent rollover in the steering process of tracked vehicle is developed according to the influencing factors of steering safety. This control strategy comes from the dynamic analysis of the steering process. Therefore, we can get the relationship between the running speed and the steering angle speed when the tracked vehicle is in the critical state of rollover. In this condition, the center speed of the tracked vehicle is called the critical speed of rollover steering. Therefore, the control objectives of the anti-rollover control strategy developed in this paper are the vehicle speed and the steering angle speed. Affected by different slope angles, different steering angular speeds corresponding to different running speeds. In the simulation of the control strategy, when the actual vehicle speed exceeds the desired vehicle speed (given by the target steering angular speed), in order to prevent the vehicle from rolling over, the controller automatically adjusts the vehicle speed under the steering target.
Comment 3: It is required to substantiate the choice of PID controller by the authors. Why was it not enough to use a PI or P controller?
Answer: Thank you very much for your valuable comments. The answer to this question is as follows: The vehicle driving control system designed in this paper adopts Fuzzy-PID control algorithm. Because the orchard tracked vehicle studied is a typical nonlinear high-order system, which adjusts the running speed of the tracks on both sides through the electro-hydraulic proportional control system. In addition, due to the complex dynamic properties between the chassis and the ground of the orchard operation vehicle, the driving conditions change constantly, and the high requirements for speed control performance. In this condition, the traditional PID control cannot make the system obtain good stability and dynamic performance, so that the Fuzzy-PID is used as the control algorithm of the driving system. Compared with traditional PI or P control, this controller has better dynamic performance, such as obviously reducing system overshoot and accelerating target response speed.
Comment 4: The authors did not present the results of experimental studies confirming the correctness of theoretical studies.
Answer: Thank you for your suggestions. The real vehicle experiments under different terrain can better test the correctness of the proposed control method, thus, building the test platform is what we were doing. However, at present, the test crawler is in the assembly stage, and the reconstruction of the test site has not been completed. During the development of the tracked vehicle, we put forward the steering safety control strategy of the hydrostatic driven tracked vehicle, and preliminarily verified our theory through computer simulation.
